# CAUSAL NEURAL NETWORKS FOR CONTINUOUS TREATMENT EFFECT ESTIMATION

## ABSTRACT

Causal inference have wide applications in medical decision-making, evaluating advertising, and voucher distribution. The exist of confounding effect makes it difficult to have an unbiased uplift estimation. Traditional methods focuses on the ordering of the problem. Little attention has been paid to the response performance, either on the evaluation metric, nor the modeling. In this work, an end-to-end multi-task deep neural network is proposed to capture the relations between the treatment propensity and the treatment effect, where the treatment can be continuous. The performance of the proposal is tested over large scale semi-synthetic and real-world data. The result shows that the proposal balances the estimation of response performance and individual treatment effect. The online environment implementation suggests the proposal can boost up the market scale and achieve 4.8% higher return over investment (ROI).

## 1 INTRODUCTION

Conditional average treatment effect (CATE), also known as causal inference, can characterize the causal effect of certain treatments on the potential outcome over the whole population as a function of measured features or covariates. Typically, samples are separated into two groups, treatment group and control group. The treatment group is given treat, such as medicines, while the control group is not. Causal inference is to estimate the treatment effect, also known as uplift, over the control group. This technique has wide applications such as identifying the best medical decisions (Glass et al. (2013)), the most effective policy to economies (Heckman & Vytlacil (2007)), the most popular item for consumers on internet platforms (Liang et al. (2016)).

One of the most significant issues encountered in practice in the causal inference problem is the biased distribution of treatment and control group. Ideally, the model should be trained over randomized control trials (RCT) data, in which treatment should be independent of features. However, the RCT experiment is usually not available or expensive to obtain in practice. Typically, confounding factors exist which affect the treatment and potential outcome at the same time in the observational data. In the internet scenario, the voucher is often provided only to inactive or new users. The active users, on the other hand, are not given the vouchers for cost-saving. This phenomenon makes it difficult to learn an unbiased model.

Various methods have been developed to alleviate the bias introduced by confounders, such as, sensitivity analyses (Rosenbaum & Rubin (1983)), instrumental variable (IV) approaches (Angrist et al. (1996)), and multi-task deep neural network (DNN) (Shi et al. (2019)). However, the sensitivity analyses can not give the point estimation. IVs can only partially identify CATE. Not to mention that most of the works limit to categorical treatment.

In some scenarios, it is sufficient to get an ordering of uplift of samples, while the response estimation is not crucial. In a recommendation system, the cost of pushing an advertise to a consumer is constant, no matter whether the consumer will purchase or not. In this case, one just needs to figure out consumers that have the greatest uplift. However, when it comes to the resource allocation problem, accurate response estimation matters. For example, with the advertise, subsidy or a coupon is sent to consumers. Naturally, the platform want to control the total budget. In this case, one not only needs to find consumers with the largest uplift, but also needs to estimate the probability of purchasing and the expected cost. In this case, the commonly used Qini coefficients and aera under uplift curve (AUUC) are not enough, since they does not describe the response estimation performance.

The main contribution of this paper are as follows:

- An end-to-end multi-task deep neural network (DNN) is proposed to estimate the effect of continuous treatment which captures the relationships between the treatment propensity, the true responses, and the treatment effect to alleviate treatment bias in an integrated manner.
- Extensive experiments carried out on semi-synthetic and real-world dataset indicate that the proposal outperforms the benchmarks in ITE estimation accuracy and uplift ranking performance.
- The proposal is implemented in an online environment of voucher distribution on the largest ride-hailing platform in mainland China. The results illustrate the effectiveness of the proposal in scaling up the market and achieving high return over investment (ROI).

This work is organized as follows. In Section 3, the problem setup and notations used in this work are presented. The proposals are presented in Section 4. Section 5 presents the experiments over semi-synthetic and real-world data, and Section 6 illustrates the online environment results. Section 7 concludes the work.

## 2 Related Work

There are in general three main approaches to estimate causal effects, two-model, single-model, and direct methods. The two-model approach fits two models, one to predict the response of the control group and the other to predict the treatment group response (Radcliffe (2007); Nassif et al. (2013)). The model is intuitive and simple. Mature techniques can be applied within this framework like random forest, XGBoost, and DNN. However, modeling the outcome separately often misses the "week" uplift signal (Radcliffe & Surry (2011); Zaniewicz & Jaroszewicz (2013)).

Single-model framework typically models a transformed target variable $Z = YT + (1 - Y)(1 - T)$ where $Y$ is the binary outcome and $T$ is the binary treatment (Jaskowski & Jaroszewicz (2012)). Following this idea, another transformation is presented in (Athey & Imbens (2015)) to relax the balance treatment assignment assumption. Single-model framework often outperforms the two-model framework. But the binary treatment and outcome assumption limits its applications.

Tree based model is typically employed to directly model the uplift (Wan et al. (2022); Wager & Athey (2018); Tang et al. (2022)). A new splitting criterion with kernel-based doubly robust estimator is proposed in (Wan et al. (2022)) to alleviate the biased treatment distribution. However, tree models have difficulties in dealing with continuous treatment.

Deep causal models have attracted increasing attentions since 2016 (Johansson et al. (2016)). A domain adaption framework is adopted in (Shalit et al. (2017); Alaa et al. (2017)). Causal representation learning is presented in (Atan et al. (2018). Generative Adversarial Nets (GAN) models are considered in (Bica et al. (2019; 2020))

Recently, randomized control trials are combined with the observational data to adjust confounders (Concato et al. (2000)), with the assumption of transportability of RCT to external observational studies (Pearl & Bareinboim (2011)). (Yu et al. (2023)) proposed a transfer learning enhanced uplift model that utilizes biased observational data to estimate the uplift of RCT data. However, the RCT data is usually expensive to obtain and much less than the observational data.

Most works in causal inference area focus on the binary (Shi et al. (2019); Tang et al. (2023; 2022); Zhong et al. (2022)) or discrete (Wan et al. (2022)) treatment case. The continuous treatment case is less studied (Wang et al. (2022); Gao et al. (2021)). In some cases, such as voucher distribution and dose selection, the continuous treatment is necessary.

## 3 Notation and Assumptions

The treatment effect estimation, also known as uplift modeling, with continuous treatment is considered in this work. Following the potential outcome framework in (Rubin (1974)), let $T$ be the 1-dim continuous treatment, $\mathbf{X}$ be the $d_X$-dim confounding variables, and $Y \in \{0, 1\}$ be the outcome of interest. As shown in Figure 1a, the population $\Omega : (\mathbf{X}, T, Y)$ is assumed to satisfy the following

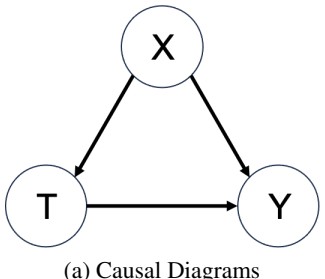

(a) Causal Diagrams

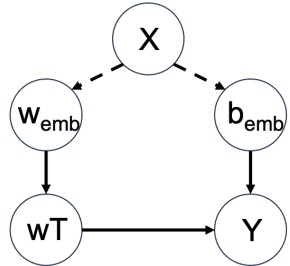

(b) Intuition: two embeddings are trained. The $b$ part indicates the base response without treatment. The $w$ part describes the treatment effect.

Figure 1: Causal Diagrams and intuition

equations:

$$Y = g(T, \mathbf{X}) + \epsilon; \quad g(T, \mathbf{X}) = h(T, \mathbf{X}) + b(\mathbf{X}); \quad T = f(\mathbf{X}) + \nu$$

where $\epsilon$, $\nu$ are white noises. The samples $\{(\mathbf{X}_i, T_i, Y_i)\}$ where $i = 1, \ldots, n$ are drawn from $\Omega$. The potential outcome under treatment $T = t$ is defined as $Y_{(t)}$. In the context of continuous treatments, the propensity score is defined by probability density function $\pi(T = t | \mathbf{X})$. The sample set of treatment $T = t$ is denoted as $N_t$ and the sample size is $n_t \triangleq ||N_t||$.

The conditional average treatment effect (CATE) is the estimation target, formally defined by

$$\theta(t, \mathbf{X}) = E[Y_{(t)} | \mathbf{X}] - E[Y_{(0)} | \mathbf{X}].$$

The following mild assumptions are made (Holland (1986); Kennedy et al. (2017)).

Assumption 1. Consistency: the outcome of any sample solely depends on its treatment.

Assumption 2. Ignorability: The potential outcomes $Y_{(T)}$ is independent of treatment $T$ given covariate $\mathbf{X}$, i.e. $(Y_{(T)} \perp T) | \mathbf{X}$

Assumption 3. Positivity: The density of treatment is bounded away from 0, i.e. $\pi(T = t | \mathbf{X}) > p_{\min} > 0$.

Assumption 4. Monotonicity: The treatment effect is assumed to be monotone, i.e. $E[Y_{(t_1)} | \mathbf{X}] \geq E[Y_{(t_2)} | \mathbf{X}]$ if $t_1 > t_2$.

Under these assumptions, we have

$$\begin{aligned} \theta(t, \mathbf{X}) \quad &= E[Y_{(t)} | \mathbf{X}] - E[Y_{(0)} | \mathbf{X}] = E[Y | T = t, \mathbf{X}] - E[Y | T = 0, \mathbf{X}] \\ &= E[g(t, \mathbf{X}) | T = t, \mathbf{X}] - E[g(t, \mathbf{X}) | T = 0, \mathbf{X}] \end{aligned}$$

Assumption 4 is typically true in the internet business scenarios. For example, users with more subsidy typically more likely to consume.

## 3.1 Evaluation

Typically, the treatment effect can not be evaluated since the counter factor is unknown. For each sample, only the outcome given one treatment or none is observed. Outcomes with other treatment are unknown. In this case, Qini coefficient and area under uplift curve (AUUC) are popular metric choices. However, the metrics only describe the ordering performance of models and not the response performance. Thus the response performance errors are introduced in this work.

For each treatment $T = t$, the uplift curve is defined as the follows. The samples from both control and treatment group $T = t$ are sorted in descending order by the estimated uplift value $\hat{\theta}(t, \mathbf{X}_i)$. The response sum of the first $k$-samples with treatment $T = t$ and without treatment are defined as

$$R(t, k) = \sum_{i \leq k} Y_i \mathbb{I}(T_i = t); \quad R(0, k) = \sum_{i \leq k} Y_i \mathbb{I}(T_i = 0),$$

where $\mathbb{I}(T_i = 0)$ is the indicator function defined as

$$\mathbb{I}(T_i = t) = \begin{cases} 1, & \text{if } T_i = t \\ 0, & \text{else.} \end{cases}$$

Define the first $k$ samples uplift score as

$$u(t, k) = [\frac{R(t, k)}{N(t, k)} - \frac{R(0, k)}{N(0, k)}][(N(t, k) + N(0, k)].$$

where $N(t, k) = \sum_{i \le k} \mathbb{I}(T_i = t)$.

Given the first to the $(n_t + n_0)$th-samples uplift score, the uplift curve can be plotted. The AUUC for treatment $t$ and the entire dataset are calculated as

$$\text{AUUC}^t = \frac{\sum_k u(t, k)}{n_t + n_0}; \text{AUUC} = \frac{\sum_t \text{AUUC}^t n_t}{\sum_t n_t}.$$

Qini coefficient is a normalization of AUUC and can be compared across different datasets. Define:

$$q(t, k) = R(t, k) - R(0, k)\frac{N(t, k)}{N(0, k)}$$

Qini coefficient for treatment $t$ and for the entire dataset are calculated as

$$\text{Qini}^t = \frac{\sum_k q(t, k)}{n_t + n_0}; \text{Qini} = \frac{\sum_t \text{Qini}^t n_t}{\sum_t n_t}.$$

To evaluate the response performance, we define the average treatment effect (ATE) error of treatment $t$ as

$$\epsilon_{\text{ATE}}^t = \text{abs}[(\frac{R(t, n_t)}{n_t} - \frac{R(0, n_0)}{n_0}) - (\frac{\sum_{i \in \mathcal{N}_t} \hat{g}(t, X_i)}{n_t} - \frac{\sum_{i \in \mathcal{N}_0} \hat{g}(0, X_i)}{n_0})]$$

where $\mathcal{N}_t$ is the set of populations with treatment $T = t$, and $\hat{g}$ the response estimate given by the model. ATE error measures the average difference between the true and estimated treatment effect in a treatment group.

Since the ground truth is known in the synthetic data, we can directly evaluate the estimate of individual treatment effect (ITE). The MAE for treatment $t$ is defined as

$$\epsilon_{\text{prob}}^t = \mathbb{E}[\text{abs}[g(t, \mathbf{X}_i) - \hat{g}(t, \mathbf{X}_i)]]$$

where $g(t, \mathbf{X}_i)$ is the true probability that $Y_i = 1$, and $\hat{g}(t, \mathbf{X}_i)$ is the estimate. The response error for treatment $t$ is defined as

$$\epsilon_{\text{Aprob}}^t = \mathbb{E}[\text{abs}[Y_i - \hat{g}(t, \mathbf{X}_i)]]$$

The ITE error for treatment $t$ is defined as

$$\epsilon_{\text{ITE}}^t = \mathbb{E}[\text{abs}[(g(t, \mathbf{X}_i) - g(0, \mathbf{X}_i)) - (\hat{g}(t, \mathbf{X}_i) - \hat{g}(0, \mathbf{X}_i))]].$$

The ATE error $\epsilon_{\text{ATE}}$, MAE $\epsilon_{\text{prob}}$, response error $\epsilon_{\text{Aprob}}$ and ITE error $\epsilon_{\text{ITE}}$ are the weighted average of $\epsilon_{\text{ATE}}^t$, $\epsilon_{\text{prob}}^t$, $\epsilon_{\text{Aprob}}^t$ and $\epsilon_{\text{ITE}}^t$, respectively.

## 4 METHOD

We present the network design in this section. As shown in Figure 2, we proposed a three-head network to deal with confounder. The features $\mathbf{X}$ and the treatment $T$ are fed to two MLPs to generate embedding. These embeddings are fed to three networks, the Monotonicity network, the Diffloss network and the logistic regression network. Three errors are weighted summed to generate the final loss. The online inference is the output of the Monotonicity network. Figure 2 shows the design when the output is categorical. When the potential output is continuous, simple modification is needed in the loss design.

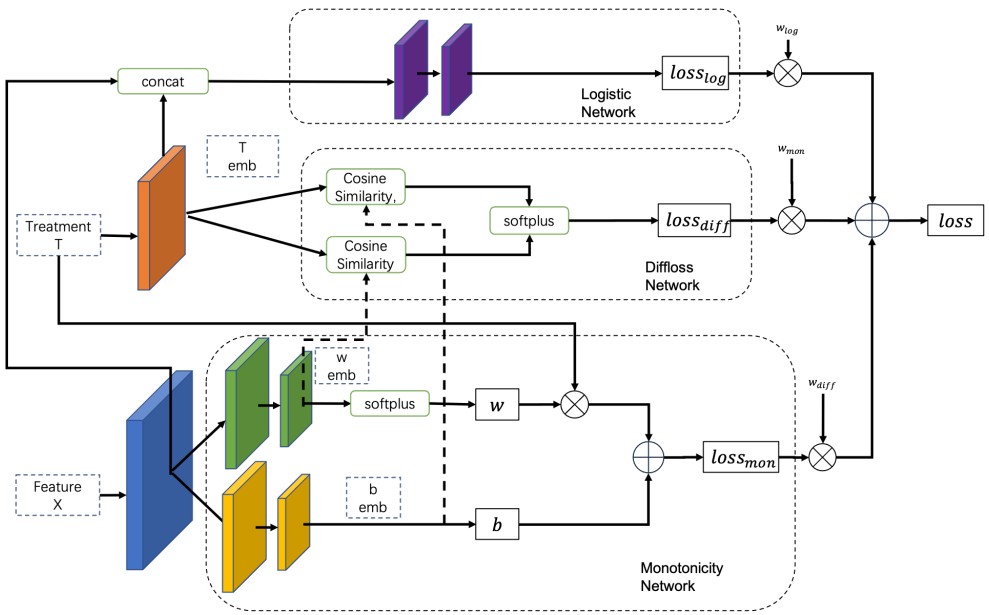

Figure 2: Method design

## 4.1 LOGISTIC NETWORK

The treatment embedding and feature embedding are concatenated and fed to the logistic network. The label is the binary outcome $Y$, and the error is cross-entropy. This network is designed to help train meaningful treatment and feature embedding. When the outcome $Y$ is continuous, the error should be replaced by mean squared error (MSE) or mean absolute error (MAE).

## 4.2 MONOTONICITY NETWORK

According to Assumption 4, the monotonicity network is designed to guarantee the monotonicity. The probability of $Y = 1$ is assumed to follow a linear equation as follows

$$\hat{P}(Y = 1) = \text{sigmoid}(w * T + b),$$

where the value of $w$ and $b$ is the output of the $w$-DNN and $b$-DNN, correspondingly. The input of these two DNNs is the feature embedding. The value $b(\mathbf{X})$ describes the potential outcome without treatment. The value $w(\mathbf{X}) * T$ describes the effect of treatment.

To guarantee the monotonicity of treatment effect, the activation function of the $w$-DNN is softmax. The loss is defined as the cross-entropy. When the potential outcome is continuous, the error should be replaced by MSE or MAE.

## 4.3 DIFFLOSS NETWORK

In the monotonicity network, the probability of $Y = 1$ is assumed to follow a linear equation with two parts. The value $b$ is assumed to be independent of treatment $T$. In the Diffloss network, the outcome of the $b$-network and $w$-network is trained to be orthogonal. The cosine similarity between the hidden state of the $b$-network ($w$-network) and the treatment embedding is calculated and denoted by $\cos_b$ ($\cos_w$).

$$\cos_b = \frac{h_b * emb_T}{||h_b|| * ||emb_T||}.$$

Thus the loss is defined as

$$\text{loss}_{\text{diff}} = -\log \frac{\exp(\gamma \cos_w)}{\exp(\gamma \cos_w) + \exp(\gamma \cos_b)}.$$

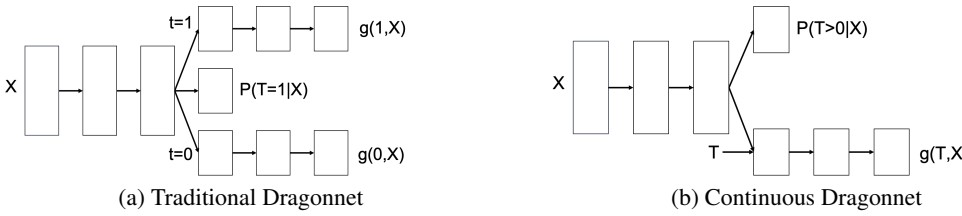

Figure 3: Dragonnet structures

Minimizing this loss means to maximize the cosine similarity between the hidden state of $w$-network and the treatment embedding while minimize the one between the hidden state of $b$-network and the treatment embedding, which makes $b$ to be independent of treatment $T$.

Given three networks, the loss is the weighted sum as follows:

$$\text{loss} = \omega_{\text{mon}}\text{loss}_{\text{mon}} + \omega_{\text{log}}\text{loss}_{\text{log}} + \omega_{\text{diff}}\text{loss}_{\text{diff}}$$

The inference is given by

$$
\begin{aligned}
\hat{\theta}(t, \mathbf{X}) \quad &= \text{sigmoid}\big(\hat{w}(t, \mathbf{X}) * t + \hat{b}(\mathbf{X})\big) - \text{sigmoid}\big(\hat{w}(0, \mathbf{X}) * 0 + \hat{b}(\mathbf{X})\big) \\
&= \text{sigmoid}\big(\hat{w}(t, \mathbf{X}) * t + \hat{b}(\mathbf{X})\big) - \text{sigmoid}\big(\hat{b}(\mathbf{X})\big)
\end{aligned}
$$

where $\hat{w}$ and $\hat{b}$ are the output of the monotonicity network.

### 4.4 Discussion

In (Rosenbaum & Rubin (1983)), a theorem of sufficiency of propensity score is stated as follows.

**Theorem 1** (Sufficiency of Propensity Score). If the average treatment effect $\theta$ is identifiable from observational data by adjusting for $\mathbf{X}$, i.e., $\theta = E[E[Y|T = t, \mathbf{X}] - E[Y|T = 0, \mathbf{X}]]$, then adjusting for the propensity score also suffices, $\theta = E[E[Y|T = t, f(\mathbf{X})] - E[Y|T = 0, f(\mathbf{X})]]$.

Motivated by this theorem, a three headed deep neural network, Dragonnet is proposed to deal with confounders when the treatment is binary in (Shi et al. (2019)). As shown in Figure 3a, one head predicts the probability that one sample gets treated, and other two heads predicts the potential outcomes when the treatment equals to 0 and 1, correspondingly. However, this structure can not fit the continuous treatment case. One naive extension of Dragonnet is shown in Figure 3b. One head is to predict the probability that one sample gets treated ($T > 0$), and the other head predicts the potential outcomes with different treatment.

This model does not guarantee the monotonicity of the treatment effect. One intuitive idea is to separate the potential outcomes into two parts. As shown in Figure 1b, one part is the base part that has nothing to do with treatments, and represents the natural outcomes without treatments. The other part is the treatment part, which tries to estimate $h(T, \mathbf{X})$ and keep the monotonicity in $T$. To make sure the first part has nothing to do with treatments, the Diffloss network is introduced to make $w$ and $b$ orthogonal. The Logistic network is to make the embedding meaningful.

## 5 Experiments

### 5.1 Pre-established causal benchmark dataset

The IBM causal inference banchmark framework is used to evaluate the performance of the model, which was developed for the 2018 Atlantic Causal Inference Conference competition data (ACIC 2018). The dataset is derived from the linked birth and infant death data (LBIDD) (MacDorman & Atkinson (1998))[1]. The treatment in the dataset is binary and the potential output is continuous. Thus the activation function of the logistic network and monotonicity network are set to none. Besides, the

---

[1]https://github.com/IBM-HRL-MLHLS/IBM-Causal-Inference-Benchmarking-Framework/tree/master/data/LBIDD

treatment effect in the data is not necessarily positive. Thus, the softplus function in the monotonicity network is removed.

Table 1: ACIC Data Results

| Method | $\epsilon_{\text{prob}}$ | $\epsilon_{\text{ITE}}$ | $\epsilon_{\text{prob}}$ All | $\epsilon_{\text{ITE}}$ All | # of params | Time |
|---|---|---|---|---|---|---|
| DragonNet | 1.3527 | 9.0548 | 2.2591 | 8.9741 | 176804 | 5.08s |
| WTB-diff | **1.3151** | **6.3749** | **1.3753** | **5.0956** | 14935 | 3.9s |

Since the outcome of this dataset is continuous, only the ITE error and MAE are presented. The proposal is compared with Dragonnet (Shi et al. (2019)). In Table 1, the performance with two settings are presented as the one in (Shi et al. (2019)). Left half of the table presents the test errors. That is to divide the dataset to training set and test set, with ratio 8:2. The model is trained over the training set and the tested over the test set. The right half of the table is to train and test models using the whole dataset. The result shows that the MAE and ITE error of the proposal are smaller than the ones of Dragonnet. The number of hyper-parameters in the models and the training time for one epoch are also smaller.

## 5.2 SEMI-SYNTHETIC DATA

The data is collected from one of the largest ride-sharing platform in mainland China, which covers from July 2022 to April 2023. Users come to the platform and submit their ride enquiry to the platform, including origins and destinations. The platform will provide estimated cost of the trip. Meanwhile, a possible subsidy is provided by the platform. Given the cost and subsidy, users decide to whether to place the order. In this work, more than 8 million enquires are collected, in which there are more than half million samples are RCT data. The data is sampled that the treated and non-treated samples are 1:1. There are more than 300 features, including the enquiry information, the historical behavior of the user, and the supply-demand information of the origin.

The semi-synthetic data is generated using the real enquires. The treatment is the discount rate. Two datasets are examined in this work. The linear dataset assumes the probability a user places an order is a linear function of the treatment and features as defined as follows:

$$\begin{cases} P(Y = 1|\mathbf{X}_i) = \text{sigmoid}(w_0 * T_i + b_i) \\ b_i = \alpha\mathbf{X}_i \end{cases}$$

where $\alpha = [\alpha_j]$ are randomly generated, and $w >> \alpha_j$ to make uplift effect significant.

The polynomial dataset assumes the probability a user places an order is a polynomial function of the treatment and features as defined as follows:

$$\begin{cases} P(Y = 1|\mathbf{X}_i) = \text{sigmoid}(w_i * T_i + b_i) \\ b_i = \alpha\mathbf{X}_i \\ w_i = w_0 + \lambda T_i + \beta\mathbf{X}_i \end{cases}$$

where $\alpha, \beta, \lambda$ are randomly generated, and $\lambda >> \alpha_j$ to make uplift effect significant.

Table 2: Linear Semi-synthetic Data Results

| Method | $\epsilon_{\text{prob}}$ | $\epsilon_{\text{ITE}}$ | AUUC | Qini | $\epsilon_{\text{ATE}}$ |
|---|---|---|---|---|---|
| Continuous DragonNet | 0.0320 | **0.0150** | **0.5216** | 0.3445 | 0.0068 |
| GCF | - | 0.0185 | 0.5106 | -0.0018 | - |
| ADMIT (modified) | 0.085 | 0.0324 | 0.4902 | 0.1343 | 0.0468 |
| XGBoost(rand) | 0.0929 | 0.0282 | 0.4273 | 0.3179 | 0.0111 |
| WTB | **0.0181** | 0.0182 | 0.5036 | 0.3518 | 0.0020 |
| WTB-diff | 0.0244 | 0.0185 | 0.4960 | **0.3519** | **0.0019** |
| WTB-diff-fft | 0.0195 | 0.0174 | 0.4891 | 0.3509 | 0.0020 |

The linear semi-synthetic data results are listed in Table 2. Continuous DragonNet is a two headed deep neural network presented in Figure 3b. Generalized Causal Forest (GCF) (Wan et al. (2022)) introduces a new splitting criterion with kernel-based doubly robust estimator. The ADMIT method

employs an importance sampling technique to deal with the biased treatment distribution (Wang et al. (2022)). The WTB model serves as an ablation study. It follows the structure in Figure 2, without the Diffloss Network and the the Logistic Network. The WTB-diff-fft model is similar to the proposal in Figure 2. The only difference is that the WTB-diff-fft model replaces the learnable treatment embedding MLP by an FFT module. All models is trained using biased observe data and tested on the RCT data, except XGBoost(rand), since it is not designed to deal with the confounders. XGBoost(rand) is a s-learner model, which uses RCT data only to fit the potential outcome, with treatment as one feature. If the sample is from the control group, then $T = 0$.

It is shown in Table 2 that the dragonnet has the smallest ITE error and greatest AUUC value. But the MAE $\epsilon_{\text{prob}}$ is large, which means the ordering of Dragonnet is good but the response estimation is not that good just as stated in (Shi et al. (2019)). On the other hand, three proposals get both MAE and ITE error small, which balances the ordering and the response estimate. GCF also performs great in the ordering. However, it does not provide the response estimate. XGBoost(rand) performs the worst, even trained with the RCT data. Comparing three proposals, it can be seen that the Diffloss and logistic network module increases the response error $\epsilon_{\text{prob}}$, since it introduce additional losses. However, the WTB-diff-fft model has smaller ITE error and ATE error.

The polynomial semi-synthetic data results are listed in Table 3. Similar to the linear case, Dragonnet is good at ordering but not that good at the response estimate. The three proposals balances two aspects. Comparing three proposals, the WTB-diff model get both MAE and ITE error small and the greatest Qini coefficients. The WTB model estimates the response well, while the WTB-diff-fft model estimates ITE well.

Table 3: Polynomial Semi-synthetic Data Results

| Method | $\epsilon_{\text{prob}}$ | $\epsilon_{\text{ITE}}$ | AUUC | Qini | $\epsilon_{\text{ATE}}$ |
|---|---|---|---|---|---|
| Continuous DragonNet | 0.0378 | **0.0227** | 0.4856 | 0.3672 | 0.0144 |
| GCF | - | 0.0346 | 0.5029 | -0.0004 | - |
| ADMIT(modified) | 0.0764 | 0.0239 | 0.4519 | 0.3499 | 0.0317 |
| XGBoost(rand) | 0.0913 | 0.0483 | 0.4833 | 0.3588 | 0.0211 |
| WTB | **0.0146** | 0.0400 | 0.4379 | 0.3703 | **0.0018** |
| WTB-diff | 0.0297 | 0.0307 | 0.5002 | **0.3707** | 0.0046 |
| WTB-diff-fft | 0.0260 | 0.0305 | **0.5032** | 0.3699 | 0.0036 |

## 5.3 REAL-WORLD DATA

In the real-world data part, the data is identical to the semi-synthetic data part, except that the label is true outcomes. Since there is no ground truth in the real-world data. Only the AUUC, Qini, ATE and average response errors are included. In Table 4, the proposals achieve the best Qini coefficients, MAE and ATE errors. Comparing three proposals, WTB-diff model balances Qini and ATE errors well. WTB model gets the best ATE error and WTB-diff-fft model get the best Qini score.

Table 4: Real-world Data Results

| Method | AUUC | Qini | $\epsilon_{\text{ATE}}$ | $\epsilon_{\text{Aprob}}$ |
|---|---|---|---|---|
| GCF | 0.5179 | -0.0001 | 0.0105 | - |
| ADMIT(modified) | 0.4986 | 0.3524 | 0.030 | 0.055 |
| XGBoost(rand) | **0.6** | 0.309 | 0.009 | 0.065 |
| WTB | 0.336 | 0.288 | **0.007** | **0.005** |
| WTB-diff | 0.525 | 0.373 | 0.008 | 0.013 |
| WTB-diff-fft | 0.51 | **0.397** | 0.015 | 0.028 |

## 5.4 EFFECT OF MODEL PARAMETERS

In order to explore the effects of different hyper-parameters of our proposed model, we evaluate it on the linear semi-synthetic dataset with two important hyper parameters, dimension of FFT and $\gamma$

in the Diffloss network. As shown in Figure 5, the AUUC and Qini coefficients dip, and the MAE and ITE error increases as the FFT dimension increases. In Figure 6, increasing in $\gamma$ decreases Qini coefficients. Thus, we choose FFT dimension as 16 and $\gamma = 1$ considering resource consumption.

Table 5: Effect of FFT dimension

| Method | $\epsilon_{prob}$ | $\epsilon_{ITE}$ | AUUC | Qini | $\epsilon_{ATE}$ |
|---|---|---|---|---|---|
| WTB-diff-fft-16 | 0.0195 | 0.0174 | 0.4891 | 0.3509 | 0.0020 |
| WTB-diff-fft-32 | 0.0264 | 0.0176 | 0.4615 | 0.2354 | 0.0019 |
| WTB-diff-fft-64 | 0.0268 | 0.0211 | 0.4865 | 0.3509 | 0.0019 |
| WTB-diff-fft-128 | 0.0261 | 0.0207 | 0.4979 | 0.3513 | 0.0019 |

Table 6: Effect of $\gamma$

| Method | $\epsilon_{prob}$ | $\epsilon_{ITE}$ | AUUC | Qini | $\epsilon_{ATE}$ |
|---|---|---|---|---|---|
| $\gamma = 1$ | 0.0264 | 0.0176 | 0.4615 | 0.3509 | 0.0019 |
| $\gamma = 2$ | 0.0242 | 0.0215 | 0.4557 | 0.2354 | 0.0019 |
| $\gamma = 10$ | 0.0173 | 0.0195 | 0.4619 | 0.2353 | 0.0020 |

# 6 IMPLEMENTATION

The proposal is employed in the largest ride-hailing platform in mainland China. The causal inference results are employed to distribute vouchers to users so the total market scale is maximized. The uplift model firstly infers query-wise elasticity, then a clustering procedure is applied to achieve consumer fairness. After that, the class-wise elasticity and budget constraints are fed to an optimization module to generate a subsidy dictionary. At a fixed interval, the order information flows back to the optimization block to update the subsidy dictionary. The experiment is carried out in six of the largest cities in mainland China over a week in August 2023. The target subsidy rate is set to 3%. Three methods are compared with a control group. The market scale of the control group is normalized to 1, and the cost is 0 since there is no subsidy in this group. The proposal achieves the largest market scale gain with the largest cost. The return over investment (ROI) is 4.8% higher than the ones of DragonNet and XGBoost.

Table 7: Online Implementation Results

| Method | Market Scale | Cost | ROI |
|---|---|---|---|
| Control | 1 | 0 | - |
| Continuous DragonNet | 1.0176 | 3.11% | 0.56 |
| XGBoost(rand) | 1.0143 | 2.49% | 0.56 |
| WTB-diff | **1.019** | **3.18%** | **0.587** |

# 7 CONCLUSION AND FUTURE WORK

In this work, the CATE estimation problem is considered with the exists of confounders. An end-to-end multi-task network is proposed to capture the relationship between treatment propensity and the treatment effect. Numerical experiments using semi-synthetic and real-world data illustrate the effectiveness of the proposal in ITE estimation accuracy and uplift ranking performance. Online implementation shows the proposal achieves greater market scale and higher ROI.

Potential future works include to find the theoretical prove of the performance and evaluate the stability of the proposal. Another interesting problem would be to find an evaluation metric to balance the uplift ranking and response estimation performance. This metric is crucial when choosing models in the practice.

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
