# OpenReview forum: "CAUSAL NEURAL NETWORKS FOR CONTINUOUS TREATMENT EFFECT ESTIMATION"
_ICLR.cc/2024/Conference — ICLR 2024 Conference Withdrawn Submission_

### Official Review · Reviewer_pLg1 · 2023-10-14

**Soundness:** 2 fair
**Presentation:** 1 poor
**Contribution:** 2 fair
**Rating:** 3
**Confidence:** 4

**Summary:**

In this paper, authors have proposed a mutlitask style deep neural network for individualised treatment effect (ITE) estimation with continuous treatments. The proposed network disentangles base response from the treatment response using consine similarity, and also guarantees the monotonicity of treatment effect. The empirical evidence is presented for the proposed method through semi-synthetic as well as real world datasets.

**Strengths:**

- paper presents a novel deep learning based approach for ITE estimation. The novelty comes from network's approach to disentangle base response and treatment response through orthognality achieved through similarity index. Additionally, the network guarantees the monotonicity of treatment effect.
- paper also presents a real world case study as the proposed method is applied to a large-scale study of ride-sharing platform which studies the effect of discounts given to customers, and shows higher return over investment as compared with other methods.

**Weaknesses:**

1. Paper lacks clarity. Introduction and related work sections are not written clearly.
- It appears authors have tried to focus on broader review of related work on treatment effect, while they were talking about continuous treatment. They did not dig into cotinuous treatment literature properly and referred only couple of references about it. There must have been one subsection on continous treatments. Authors could have considered and discussed practical application of ITE, similar to their work in online marketing etc. This makes the related work weak and also lack any literature gap statement.
- Introduction also discusses the motivation and contribution of the paper at high level. For example, authors did not tell why the proposed method is needed? Moreover, they say they proposed a multitask DNN to addresss selection bias, however, isn't everyone nowadays proposing such DNNs? Authors must give readers an idea why you are developing a new method and how the method is working.
- Discussion in '4.4 Discussion' is not clear to me. Here, you should have discussed how different compoents working together and how is the confounding addressed in the observational data. Discussion on DragonNet seems unnecessary at this point.

2. Baselines are selected poorly.
You selected only 4 baselines. Off which DragonNet is old XGBoost is not specific for solving the continous treatment problem. So, there are only two baselines. Why other methods like specific for continuous treatment are not considered, such as VCNet, VCNet_TR [1], and TransEE[2]?

References:
[1] Lizhen Nie, Mao Ye, Qiang Liu, and Dan Nicolae. Vcnet and functional targeted regularization for learning causal effects of continuous treatments. arXiv preprint arXiv:2103.07861, 2021.
[2] YiFan Zhang, Hanlin Zhang, Zachary Chase Lipton, Li Erran Li, and Eric Xing. Exploring transformer backbones for heterogeneous treatment effect estimation. In NeurIPS ML Safety Workshop, 2022

**Questions:**

1. Causal diagram 1(b) looks somewhat wrong to me. From the diagram, it appears as you decomposed X into two latent factors for instrument variable and adjustment variable. Then where did the confounding effect go? If you dropped the confounding by learning such latent factors then that's loss of information. Isn't it? Please clearly explain how is the proposed method addressing confounding bias.
2. What do you mean by online inference in 'The online inference is the output of the Monotonicity network'?
3. Method 'WTB-diff' is used in Table 1 without defining first. Moreover, why only one baseline is used?

---

### Official Review · Reviewer_iNGX · 2023-10-24

**Soundness:** 3 good
**Presentation:** 2 fair
**Contribution:** 2 fair
**Rating:** 5
**Confidence:** 4

**Summary:**

The paper proposes a multi-task deep neural network for estimating the causal effects under the situation when the treatment is continuous. The network architecture is designed so that the relationship (e.g., monotonicity and orthogonality) between the treatment and responses are considered. The authors conduct experiments on large-scale semi-synthetic and real-world datasets to show that the proposed model can not only identify the subject with the largest uplift but also accurately estimate the response performance.

**Strengths:**

•	The motivation of designing the network architecture is clearly elaborated and illustrated in the causal graph.

•	Assumption 4 seems to be novel and make sense in most situations.

•	The proposed framework is tested in an online setting, which demonstrate its potential value in industrial applications.

**Weaknesses:**

Although Assumption 4 seems to make sense, it is not fully demonstrated that the monotonicity network mainly contributes to the performance of the proposed DNN-based model. The idea of alleviating treatment bias by making the treatment propensity and the treatment effect orthogonal has already been explored in some previous studies, such as Hatt and Feuerriegel [1], and is proved effective when compared to many other benchmarks including Dragonnet. I encourage the authors to at least discuss the similarities and differences between the previous studies and this paper in the Related Work section. Also, the paper lacks comparison with other multi-task DNN-based benchmarks such as TARNet [2], CMDE [3], and SCI [4]. A survey of deep causal models and their applications can also be found in [5].

Minor changes:

•	The paper seems to use an older version of ICLR template (ICLR 2023 instead of 2024).

•	In abstract, the “existence” of confounding effect…, either on the evaluation metric, “or” the modeling…

•	There are several typos and grammar errors in the manuscript. I strongly encourage the authors to carefully go over the paper prior to its potential acceptance.

References:

[1]. Hatt, T., & Feuerriegel, S. (2021, October). Estimating average treatment effects via orthogonal regularization. In Proceedings of the 30th ACM International Conference on Information & Knowledge Management (pp. 680-689).

[2]. Shalit, U., Johansson, F. D., & Sontag, D. (2017, July). Estimating individual treatment effect: generalization bounds and algorithms. In International conference on machine learning (pp. 3076-3085). PMLR.

[3]. Jiang, Z., Hou, Z., Liu, Y., Ren, Y., Li, K., & Carlson, D. (2023). Estimating Causal Effects using a Multi-task Deep Ensemble. arXiv preprint arXiv:2301.11351.

[4]. Yao, L., Li, Y., Li, S., Huai, M., Gao, J., & Zhang, A. (2021, October). SCI: subspace learning based counterfactual inference for individual treatment effect estimation. In Proceedings of the 30th ACM International Conference on Information & Knowledge Management (pp. 3583-3587).

[5]. Li, Z., Zhu, Z., Guo, X., Zheng, S., Guo, Z., Qiang, S., & Zhao, Y. (2023). A Survey of Deep Causal Models and Their Industrial Applications.

**Questions:**

•	Although the model can be adapted to the case with continuous outcomes, have the authors conducted any experiments containing continuous $Y$? From my perspective, using a continuous $T$ along with a binary $Y$ under the scope of this paper seems to be a limitation as the authors mention the case where accurate response estimation is important, such as the resource allocation problem.

•	Can authors do some ablation study (e.g., by fine-tuning $w_{\text{mon}}$, $w_{\text{log}}$, and $w_{\text{diff}}$) to investigate which term in the loss function plays the most important role in alleviating treatment bias?

---

### Official Review · Reviewer_LXVg · 2023-10-27

**Soundness:** 1 poor
**Presentation:** 1 poor
**Contribution:** 1 poor
**Rating:** 3
**Confidence:** 4

**Summary:**

This paper proposes a three-head network for predicting the counterfactual outcome. The network adopts cosine similarity to make embedding b independent of treatment T, while preserving the correlation between embedding w and T.

**Strengths:**

Continuous treatment effect estimation is a significant problem in causal inference.

**Weaknesses:**

- **The statement "The following mild assumptions are made (Holland (1986); Kennedy et al. (2017))" does not correspond to reality. The Monotonicity Assumption is not mentioned in (Holland (1986); Kennedy et al. (2017)), and the Consistency Assumption also differs from the one proposed by Kennedy et al. (2017). Furthermore, there are many other statements in this paper that do not match reality.**

- The writing needs significant improvement. The survey on relevant literature is not comprehensive.

- The statement "Traditional methods focus on the ordering of the problem. Little attention has been paid to the response performance, neither on the evaluation metric, nor the modeling" is confusing. In both binary and continuous settings, researchers have recently proposed numerous estimators to predict CATE or ITE, such as Johansson et al. (2016), Shalit et al. (2017), Bica et al. (2019; 2020), Schwab et al. (2020) and Nie et al. (2021). Recent deep methods typically use PEHE or MSE on CATE to evaluate the response performance, and none of them focus on the ordering of the problem. This statement contradicts my understanding.

- The statement "Not to mention that most of the works limit to categorical treatment" is incorrect. Recently, many machine learning IV algorithms have been proposed to identify CATE and are not limited to categorical treatment, such as NPIV, DeepIV, KernelIV, DualIV, FastIV, DFIV, CBIV, AGMM, and DeepGMM.  **The survey on relevant literature is not comprehensive.**

- The statement "Most works in causal inference area focus on the binary or discrete treatment case. The continuous treatment case is less studied" is incorrect. Many studies have been conducted on estimating continuous treatment effects, particularly in predicting average dose-response function (ADRF) or average dose-response curve (ADRC) to identify causal effects. Some notable works in this area include (Robins et al., 2000), (Hirano & Imbens, 2004), (Imai et al., 2004), (Flores et al., 2007), (Flores et al., 2012), (Kennedy et al., 2017), (Wilson et al., 2018), (Kallus et al., 2019), (Colangelo et al., 2020), (Sharma et al., 2020), (Klosin, 2021), (Linton & Zhang, 2021), (Zhang et al., 2022), and many others. **The survey on relevant literature is not comprehensive.**

- That statement "Conditional average treatment effect (CATE) is known as causal inference" is incorrect. Conditional average treatment effect (CATE) is not synonymous with causal inference. CATE refers to the average effect of a treatment (intervention) on an outcome under a specific condition. Causal inference, on the other hand, involves analyzing causal relationships to determine the effect of a treatment (intervention) on an outcome of interest. CATE is one method used in causal inference, but they are not identical concepts.

- **Algorithm**: The feature X is a common cause of treatment T and outcome Y, which could potentially confound the causal effect of T on Y. Although this paper uses lossdiff to make embedding b independent of treatment T, the embedding factor w may remain confounders of the causal effect of T on Y. How to address the confounding bias from w?

- **Experiments**: Due to the incomplete literature review, the paper lacks a comprehensive evaluation and comparison with existing methods, such as GPS (Imbens, 2000; Fong, 2018), DRNet (Schwab et al., 2020), SCIGAN (Bica et al., 2020), VCNet (Nie et al., 2021), making it difficult to assess the practical contribution of the proposed approach.

**Questions:**

See Above.

**Details Of Ethics Concerns:**

The statement "The following mild assumptions are made (Holland (1986); Kennedy et al. (2017))" does not correspond to reality. The Monotonicity Assumption is not mentioned in (Holland (1986); Kennedy et al. (2017)), and the Consistency Assumption also differs from the one proposed by Kennedy et al. (2017). Furthermore, there are many other statements in this paper that do not match reality.

---

### Official Review · Reviewer_5vkR · 2023-11-03

**Soundness:** 2 fair
**Presentation:** 2 fair
**Contribution:** 2 fair
**Rating:** 3
**Confidence:** 4

**Summary:**

This paper presents a neural network for evaluating effect of continuous treatment. Semi-synthetic and read-world data were used to evaluate the proposed method.

**Strengths:**

+ The proposed method seems with practical values based on the numbers in Table 7. However, I do not have a concrete idea on what/how had been done in the application from reading section 6.

**Weaknesses:**

- The writing of the paper has large room to improve, not up to the standard of typical NeurIPS papers. Many parts of the paper are difficult to follow, for example their definition of ATE error. It was stated by the authors that ATE measures the average difference between the “true” and estimated treatment effect in a treatment group. But in their experiment with real-world datasets where there is no available ground truth, somehow, the ATE errors were computed (Table 4). n_t, n_0 appeared in the top half of page 4 are not defined. It is not clear the T in 3.1 is continuous, binary, or categorical? The definition of u(t,k) is problematic when N(t,k) or N(0,k) is 0, which is very likely to happen when T is continuous or categorial (many categories), or k is small. There is lack of discussion on motivation and practical interpretation of the derived metrics. I was not able to find the definition of the softplus appearing in Figure 2. Is it the SoftPlus activation function, smooth approximation to the ReLU function, as implemented in PyTorch? WTB-diff is not introduced before appearing in Table 1. I guessed it represents the design illustrated in Figure 2, which is THE approach proposed in the paper.

- Their design (Figure 2) seems problematic to me with unnecessary redundancy. Both their logistic network and monotonicity network do the same classification. Why both are needed? Importantly, based on the experimental results, I do not think their design worked. In most of the experiments with WTB included (Table 2-3), WTB had the best performance while not the one that they championed, illustrated in Figure 2.

- It is not straightforward that maximizing/minimizing the cosine similarity between hidden state of networks could lead to b that is independent of T. I do not think this is the case. Rigorous proof or more discussion is needed.

**Questions:**

Refer to the list of weaknesses

---

### Official Review · Reviewer_QMLA · 2023-11-05

**Soundness:** 2 fair
**Presentation:** 2 fair
**Contribution:** 2 fair
**Rating:** 3
**Confidence:** 4

**Summary:**

The paper proposes an end-to-end DNN to estimate the effect of a continuous treatment and tests the method on semi-synthetic and real world data with existing benchmark methods.

**Strengths:**

Originality: The three component architecture is new and the model is generalizable to continuous treatment as well.

Quality: The experiments contain semi-synthetic and real world data. The method is also deployed in production.

**Weaknesses:**

The main weakness comes from the confusing writing.

1. CATE and ITE are interchangeably used. You should stick to one of them.
2. I find it confusing to introduce the evaluation metrics even before detailing the proposed method. Also, if you are using semi-synthetic data why not just use some simple metrics like the mse between true CATE and your estimated CATE.
3. In the method section, no architecture detail is given, like what are the dimensions of the embedding and hidden layers.
4. Maybe I misunderstand, but it seems that both $w$ and $b$ are vectors so how do you define sigmoid of $w*T+b$ as the probability of getting treated? $T$ here is a scalar right?
5. How do you ensure monotonicity if $T$ is continuous? I find the extension to continuous discussion very light.
6. What are the weights used to aggregate the three losses?
7. Why do we want to maximize the cosine similarity between $w$ and $T$ embeddings?
8. I think to facilitate a fair comparison, you should state how those methods being compared are implemented.

**Questions:**

See above. Also:

1. The first sentence of the paper is really not correct.
2. Typo second to last line on first page: aera-> area.

---

### Meta-Review · Area_Chair_G1UD · 2023-12-06

**Metareview:**

The reviewers raised many concerns about the work from clarity issues to misrepresenting some of the existing works and assumptions within those. The authors did not rebut any of the issues. All reviewers recommend rejection of the manuscript in its current form.

**Justification For Why Not Higher Score:**

The authors did not submit a rebuttal to address any of the reviewer concerns.

**Justification For Why Not Lower Score:**

N/A

---

### Decision · Program_Chairs · 2024-01-16

Reject